# A Scoping Review of the Prevalence of Eating Disorders in Spain

**DOI:** 10.3390/nu16101513

**Published:** 2024-05-17

**Authors:** Néstor Benítez Brito, Francisco Moreno Redondo, Berta Pinto Robayna, Jesús De las Heras Roge, Yolanda Ramallo Fariña, Carlos Diaz Romero

**Affiliations:** 1Nutrition and Bromatology Area, Department of Chemical Engineering and Pharmaceutical Technology, Faculty of Pharmacy, University of La Laguna, 38200 San Cristóbal de La Laguna, Spain; fran.mr091@gmail.com (F.M.R.); extbpintoro@ull.edu.es (B.P.R.); jherasro@ull.edu.es (J.D.l.H.R.); cdiaz@ull.edu.es (C.D.R.); 2Nutrition, Health and Food Research Group (NAYS), University of La Laguna, 38200 San Cristóbal de La Laguna, Spain; 3Canary Islands Health Research Institute Foundation (FIISC), 38004 Santa Cruz de Tenerife, Spain; yolanda.ramallofarina@sescs.es; 4Network for Research on Chronicity, Primary Care, and Health Promotion (RICAPPS), 38004 Santa Cruz de Tenerife, Spain

**Keywords:** eating disorders, eating and food ingestion disorders, anorexia nervosa, bulimia, binge eating disorder

## Abstract

Introduction: Eating disorders (EDs) are mental health illnesses with a multifactorial origin. At present, no review of indexed publications studying their prevalence in Spain is available. Material and methods: A scoping review (PROSPERO -CRD42019140884-) was carried out through systematic searches (MEDLINE, EMBASE and PsycINFO) until January 2022. Papers published in Spanish/English analysing the prevalence of EDs in Spain (population < 65 years) were selected. Results: A total of 766 articles were identified (186 eliminated as duplicates). A total of 580 articles were analysed on the basis of title and abstract, and 67 articles were selected for full-text analysis. A total of 37 studies analysed the prevalence of EDs in Spain. Conclusions: This is the first scoping review to analyse the prevalence of EDs in Spain. Puberty and adolescence are the most extensively studied stages. There is a high heterogeneity in the use of ED screening tools and a paucity of information on diagnostic tools.

## 1. Introduction

Eating disorders (EDs) are mental health illnesses with medical, nutritional, psychiatric and family complications [1]. Their origin is multifactorial, involving genetic, biological, psychological and socio-cultural factors.

The diagnostic criteria for these diseases are described in the Diagnostic and Statistical Manual of Mental Disorders in its fifth revision (DSM-5) [2]. This edition is characterised by a new name for all these illnesses (EDs) and the partial modification of certain diagnostic criteria for anorexia nervosa (AN) and bulimia nervosa (BN). In addition, new pathologies have been added, such as binge eating disorder, pica, rumination disorder, avoidant restrictive food intake disorder (ARFID) and other specified feeding or eating disorders (OSFEDs) [2]. The changes introduced in the criteria, as well as the new pathologies, arise from the need to specify new diagnoses and establish more specific treatments, thus improving the possibility of knowing the diagnostic distribution of EDs.

Furthermore, these diseases are also classified within the eleventh revision of the International Classification of Diseases (ICD-11) [3]. However, DSM-5 coding is used to a greater extent, as it allows for more specific diagnoses of EDs [4].

In general, among the mental disorders described in the literature, EDs have the highest morbidity and mortality rate [5], with age at diagnosis being the greatest predictor of mortality in these patients [6]. In particular, AN is the mental disorder with the highest mortality rate among all disorders described to date [7]. In recent decades, an increasing prevalence has been observed among young people and adults in developed countries, which has increased due to the coronavirus disease 2019 (COVID-19) pandemic [8]. However, despite its high morbidity and mortality, prevalence data on EDs in Spain are unknown. This fact justifies the importance and interest of knowing the publications made to date on the prevalence of EDs in the Spanish population.

## 2. Methodology

A scoping review was conducted through the literature, in accordance with the methodology developed by the Cochrane Collaboration [9]. The protocol was initially registered in PROSPERO (CRD42019140884). The following databases were systematically searched for articles published up to 2 January 2022: MEDLINE, EMBASE and PsycINFO.

The search strategy used was initially designed for MEDLINE (OvidSP), combining controlled vocabulary together with free text terms around the terms: “Feeding and Eating Disorders” or “Anorexia” or “Bulimia” or “Binge-Eating Disorder” or “Night Eating Syndrome” or “Osfed” or “Other Specified Feeding and Eating Disorder” or “Eating Disorder Not Otherwise Specified” or “EDNOS” or “Pica” or “Rumination Syndrome” or “Avoidant Restrictive Food Intake Disorder” or “Eating Disorder”. This strategy was subsequently adapted to the other databases consulted.

The bibliographic references obtained from each database were imported into a file in Reference Manager Edition Version 10 (Thomson Scientific, Philadelphia, PA, USA) to eliminate duplicate references. This file was then exported to a Microsoft Excel 2013 sheet (Microsoft Corporation, Redmond, WA, USA) where the removal of duplicate references could be completed and study selection was then performed.

The search for published studies was complemented by a manual examination of the bibliographic lists of the articles included. We selected papers published in Spanish and English that analysed the prevalence of EDs in the population under 65 years of age in Spain. We excluded those studies that focused on populations with specific pathologies (obesity, diabetes mellitus, cerebral palsy and intestinal diseases, among others) and specific study populations in athletes.

Two reviewers independently and in parallel peer-selected the studies by reading the titles and abstracts located through the literature search. Articles selected as relevant were analysed in full text by the two reviewers independently, who classified them as included or excluded according to the specified selection criteria. The reviewers contrasted their opinions and, where there were doubts or discrepancies, these were discussed for resolution. Where there was no consensus, another reviewer was consulted and, where necessary, a fourth reviewer was consulted. Discussions and agreements were documented.

Data extraction from the selected studies was carried out using an Excel spreadsheet (Microsoft^®^) designed ad hoc. In summary, the data extracted were those related to the identification of the study (authors, date of publication, community where the study was conducted, funding, etc.), the design and methodology (objective, design, sample characteristics, screening and assessment instruments, type of analysis, perspective, sources of information used, etc.) and the results of the study, with special attention to its variability.

## 3. Results

A total of 766 articles were identified in the different databases as shown in the flow chart (Figure 1). Of the total number of included studies, 186 were eliminated due to duplicates, resulting in a total of 580 articles for analysis based on reading the title and abstract. Sixty-seven articles were selected for full-text analysis, with 37 articles meeting the selection criteria (30 studies providing point prevalence data and 6 period studies). The reasons for exclusion are outlined in Table 1.

The selected studies have a highly variable sample size ranging from 175 to 4334 participants. About 36% of the studies had <500 participants; 36% included between 500 and 1000 participants and 26% included more than 1000 participants. Therefore, most of the articles selected for this review had a sample size of less than 1000 participants.

In general, the studies mostly included both sexes (10 were exclusively female). The study ages were highly heterogeneous, with analyses ranging in age from 9 to 44 years, although the range of 12 to 20 years of age was more frequent. The sample characteristics, measurements and prevalence values reported for each study included are shown in Table 2 (point prevalence) [10,11,12,13,14,15,16,17,18,19,20,21,22,23,24,25,26,27,28,29,30,31,32,33,34,35,36,37,38,39] and Table 3 (period prevalence) [40,41,42,43,44,45,46].

There is a high heterogeneity in the tools used for both screening and diagnosis of EDs. The most commonly used tools for screening are the Eating Attitudes Test (EAT) (EAT-40) [47], the Eating Disorder Inventory (EDI) [48] and the Bulimic Investigatory Test Edinburgh (BITE) [49], while for diagnosis, the Structured Clinical Interview with the Eating Disorders Examination (EDE) [50] and the Schedules for Clinical Assessment in Neuropsychiatry (SCAN) [51] questionnaires are used. 

Almost half of the studies do not report on the diagnostic tool used in conjunction with the clinical interview. Most diagnoses have been established following the Diagnostic and Statistical Manual of Mental Disorders, 4th edition (DSM-IV).

The main autonomous communities where most research has been carried out on the prevalence of EDs in Spain are Catalonia and Madrid.

## 4. Discussion

The study carried out is presented as the first scoping review carried out in Spain on the prevalence of EDs. This research has analysed and synthesised the literature published in this field in different databases, emphasising the importance of this type of study. However, it is noteworthy that, to date, no similar review has been carried out, especially considering the relevance of these pathologies due to their high morbidity and mortality. This will make it easier to undertake further studies on the prevalence of this disease, allowing the problem to be better understood.

The first prevalence study identified in this work was carried out in 1997 [40], although previous studies were published in non-indexed journals [52,53]. However, since that date, the study of these pathologies has increased in Spain, in line with other studies published in recent decades in other countries, especially in developed countries [54,55]. Even so, it is important to highlight the great variability of most of the studies identified in many respects.

On the one hand, there are differences in the methodology and/or screening tools used that hinder the diagnostic approach, as well as their prevalence. In addition, there are considerable differences in the population sectors analysed. In general, the information available is outdated with respect to the latest diagnostic criteria, as no study has been found that follows the current criteria proposed by the DSM-5. It should be noted that this situation does not only occur in Spain, although other countries are beginning to carry out studies with these revised criteria [56,57].

In this sense, the literature analysed for this review shows the enormous heterogeneity used in the diagnosis and classification of EDs, with numerous questionnaires and validated tools for the identification of EDs.

A total of 34 studies include risk prevalence results, in which up to ten different screening tools were identified, with the EAT [47], in its extended version, being the most widely used (Figure 2).

The most commonly applied screening tests for ED risk are the EAT-40 [47], the BITE [49] and the Sick, Control, One, Fat, Food Questionnaire (SCOFF) [58]. However, despite being the most widely used, the variability in the different cut-off points used to establish risk stands out, especially in the different versions of the EAT [47].

Regarding the diagnostic criteria, no results were obtained using the DSM-5 version, which shows a decrease in cases classified as EDNOS by the DSM-IV-TR [59] and an increase in specificity and selectivity obtained by applying the new criteria (DSM-5) [2]. In fact, the most widely used criteria have been made through the application of DSM-IV [59]. In this regard, several authors have confirmed the great impact of the change in criteria proposed in the DSM-5 on the results [18]. Fairburn and Cooper et al. [60] found surprising results, with a decrease in EDNOS (from 53% to 25% of total ED) and a considerable increase in cases classified as AN (from 8% to 29% of total ED), while BN remained stable (39%). This confirms the idea that studies in the Spanish population that take into account these new criteria are needed in order to update the data adequately. In the present review, we cannot analyse the possible differences observed since, as mentioned above, no study following the DSM-5 criteria is available [2]. On the other hand, no differences are observed when comparing the results obtained using DSM-III or DSM-IV; in both cases, most of the subjects have been diagnosed with EDNOS, with a low percentage of cases being classified as one of the full forms of the diseases (AN, BN or BED) included in this chapter of the manual.

Only 48% of the total selected studies report on diagnosis; twelve studies used a cross-sectional methodology for point prevalence and six studies used period prevalence to analyse cohorts of participants. In addition, few studies go as far as analysing false negatives by analysing those participants who scored negative in the risk screening.

With regard to the nature of the participants, it should be noted that EDs are pathologies that have been commonly and repeatedly associated with the female sex. The results found are in line with these observations, with a clearly higher prevalence of both EDs in general, and of each of them in particular, in the female sex compared to that observed in the male sex. This would explain why some of the studies only use a female population. Although it may imply a certain bias in the results, in this review, out of the total of 37 included studies, ten of them only included a female population [11,15,17,23,24,27,31,35,36,41]. Some authors consider that in early stages, such as adolescence, these differences tend to become blurred, considering it a mistake to eliminate male individuals in the prevalence of these pathologies [30]. This may be of relevance since many of the studies included in this review were conducted on adolescent subjects. It is also worth noting that when following the single-phase protocol, i.e., where only the application of the screening questionnaires is included, the prevalence obtained was higher than when applying the two-phase protocol.

The application of two-phase studies also has discrepancies. Some authors explain a first phase for screening and a second phase for diagnosis [43]. Other authors, however, establish the two phases using other procedures: a first phase for screening (without diagnosis) and a second phase to assess the occurrence of new cases after a certain time (usually 24 months later) [46]. Although both cases are valid, it becomes difficult to homogenise studies to assess the different prevalences.

The results have been obtained in the Spanish population of different autonomous communities, although the provinces where most studies have been carried out are Catalonia and Madrid. However, the internal variability of the population within Spain does not seem to be very high, so it can be considered a homogeneous population in terms of its origin and socio-cultural influence. It can be deduced that the differences observed are mainly due to the variability in the methodology used, as mentioned above.

The present review is limited in time due to a delay in analyses during the COVID-19 pandemic [61]. Also, certain articles identified could not be assessed because they were older, non-indexed publications and not fully accessible in online format. Nevertheless, it serves as a precedent to continue developing further prevalence studies on EDs in Spain and to continue towards a more precise analysis through a systematic review using the updated criteria (DSM-5). 

## 5. Conclusions

The present work symbolises the first scoping review carried out in Spain on the prevalence of EDs. A total of 37 studies have been identified that analyse both the risk and the occurrence of these diseases. The ages of puberty and adolescence are the most widely studied. However, there is a high heterogeneity in the use of ED screening tools and a paucity of information on diagnostic tools. More research is needed to analyse not only national prevalence using updated criteria (DSM-5) but also the economic burden of ED treatment, as well as cost-effectiveness analysis of different treatments.

## Figures and Tables

**Figure 1 nutrients-16-01513-f001:**
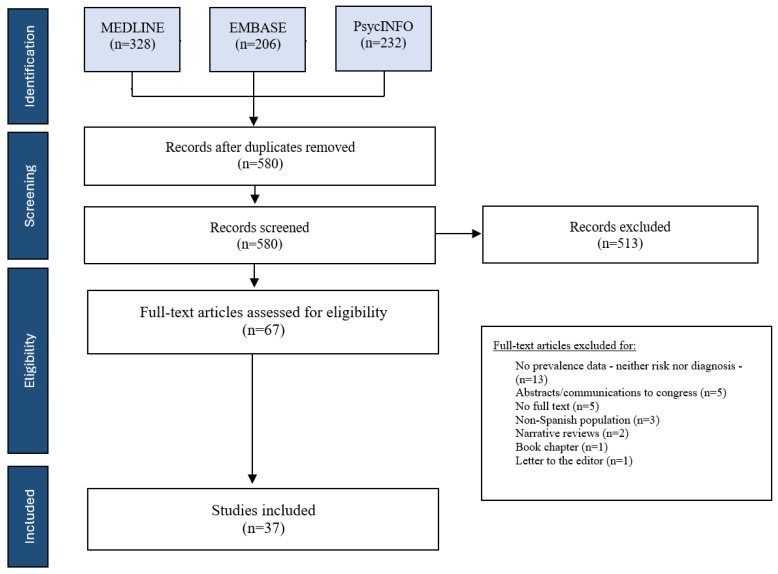
Identification of studies included in the analysis.

**Figure 2 nutrients-16-01513-f002:**
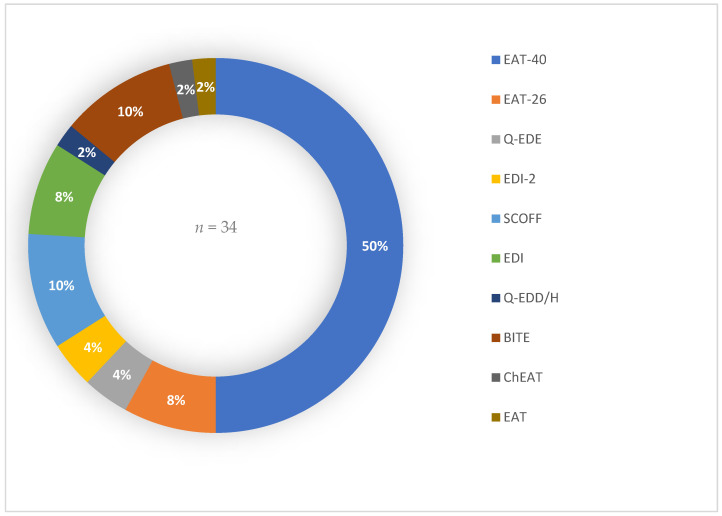
Percentage distribution of screening tools used in the different studies identified. EAT: Eating Attitude Test; Q-EDE: Questionnaire for Eating Disorder Diagnoses; EDI: Eating Disorder Inventory; SCOFF: Sick, Control, One, Fat, Food Questionnaire; Q-EDD/H: Questionnaire for Eating Disorder Diagnoses for students; BITE: Bulimic Investigatory Test Edinburgh; ChEAT: Children Eating Attitudes Test.

**Table 1 nutrients-16-01513-t001:** List of articles excluded by reason in full-text analysis.

Study (ID)	Year	Reason for Exclusion
Serra, R., et al. (ID 40)	2019	No data on prevalence, risk or diagnosis are provided.
Larrañaga, A., et al. (ID 191)	2012	No data on prevalence, risk or diagnosis are provided.
Castro Fornieles, J., et al. (ID 211)	2010	No data on prevalence, risk or diagnosis are provided.
Lahortiga Ramos, F., et al. (ID 265)	2005	No data on prevalence, risk or diagnosis are provided.
Esparó, G., et al. (ID 269)	2004	No data on prevalence, risk or diagnosis are provided.
Paniagua Repetto, H., et al. (ID 274)	2003	No data on prevalence, risk or diagnosis are provided.
Lameiras Fernández, M., et al. (ID 282)	2002	No data on prevalence, risk or diagnosis are provided.
Otero Rodríguez, J., et al. (ID 289)	2002	No data on prevalence, risk or diagnosis are provided.
Moreno Redondo, F.J., et al. (ID 357)	2019	Congress abstracts/communications.
Ruíz-Lázaro, P.M., et al. (ID 388)	2014	Congress abstracts/communications.
Lázaro, Y., et al. (ID 391)	2015	Congress abstracts/communications.
Ruíz-Lázaro, P.M., et al. (ID 413)	2013	Congress abstracts/communications.
Larrañaga, A., et al. (ID 441)	2012	No data on prevalence, risk or diagnosis are provided.
Preti, A., et al. (ID 451)	2009	Non-Spanish population.
Nuño Gutiérrez, B.L., et al. (ID 453)	2009	Non-Spanish population.
Julián Viñals, R., et al. (ID 492)	2002	Letter to the editor.
Toro, J., et al. (ID 503)	2000	Narrative review.
Ruíz-Lázaro, P.M. (ID 513)	1998	Not available in full text.
Ruíz Lázaro, P.M. (ID 527)	2001	Not available in full text.
Ballester Arnal, R., et al. (ID 570)	2000	No data on prevalence, risk or diagnosis are provided.
Peláez Fernández, M.A., et al. (ID 590)	2007	Congress abstracts/communications.
Norris, M.L., et al. (ID 592)	2011	Book chapter.
Valero Solis, S., et al. (ID 608)	2019	No data on prevalence, risk or diagnosis are provided.
Marchi, M., et al. (ID 635)	1989	Non-Spanish population.
Jiménez, M., et al. (ID 638)	2004	No data on prevalence, risk or diagnosis are provided.
Ruíz-Lázaro, P.M., et al. (ID 641)	2001	Not available in full text.
Ruíz-Lázaro, P.M., et al. (ID 663)	2005	Not available in full text.
Peláez Fernández, M., et al. (ID 671)	2005	Narrative review.
Gómez-Peresmitré, G., et al. (ID 708)	2000	No data on prevalence, risk or diagnosis are provided.
Vega Alonso, A.T., et al. (ID 710)	2001	Not available in full text.

**Table 2 nutrients-16-01513-t002:** Description of studies analysing the point prevalence of eating disorders in Spain.

Study (ID)	Year	Region	Age	Sex	Type of Study	Tools for Screening	Tools for Diagnosis	*n*	Total N Prevalence According to Tools	%	Total N Prevalence According to Diagnosis	%	*n* Anorexia	%	*n* Bulimia	%	*n* EDNOS	%	*n* Binge Eating Disorder	%
Canals, J. et al. (ID 318) [10]	1997	Tarragona	13–14	Both	Cross- sectional	EAT-40	N/A	⚥ 515	⚥ 52	♀ 12.4 ♂ 8.3 ⚥ 10.10	N/A	N/A	N/A	N/A	N/A	N/A	N/A	N/A	N/A	N/A
P.M.Ruiz, et al. (ID 594) [11]	1998	Zaragoza	12–18	Female	Cross- sectional	EAT-40	Interview semi-structured (SCAN) ^A,B^	♀ 2193	♀ 358	♀ 16.32	♀ 99	♀ 4.51	♀ 3 ^A^ ♀ 1 ^B^	♀ 0.14 ^A^ ♀ 0.045 ^B^	♀ 12 ^A^ ♀ 1 ^B^	♀ 0.55 ^A^ ♀0.045 ^B^	♀ 84 ^A^ ♀ 97 ^B^	♀ 3.83 ^A^ ♀ 97.8 ^B^	N/A	N/A
Sáiz Martínez, P.A.et al. (ID 507) [12]	1999	Asturias	13–21	Both	Cross- sectional	EDI	N/A	⚥ 816	⚥ 72	♀ 7.7 ♂ 1.1 ⚥ 8.80	N/A	N/A	N/A	N/A	N/A	N/A	N/A	N/A	N/A	N/A
Morandé, G., et al. (ID 309) [13]	1999	Madrid	15 ± 0.98	Both	Cross- sectional	EDI *	Clinical interview ^C^	⚥ 1281	⚥ 293	♀ 31.21 ♂12.02 ⚥22.87	⚥ 39	♀ 4.69 ♂ 0.90 ⚥ 3.04	⚥ 5	⚥ 0.39	⚥ 11	⚥ 0.86	⚥ 23	⚥ 1.8	N/A	N/A
Martínez Martínez, A., et al. (ID 505) [14]	2000	Asturias	14–22	Both	Cross- sectional	EAT-26	N/A	⚥ 860	⚥ 63	♀ 12.8 ♂ 1.8 ⚥ 7.33	N/A	N/A	N/A	N/A	N/A	N/A	N/A	N/A	N/A	N/A
Pérez-Gaspar, M., et al. (ID 504) [15]	2000	Navarra	12–21	Female	Cross- sectional	EAT-40	Interview semi-structured (EDI) ^A^	♀ 2862	♀ 319	♀ 11.10	119	♀ 4.16	♀ 9	♀ 0.31	♀ 22	♀ 0.77	♀ 88	♀ 3.07	N/A	N/A
Rivas, T., et al. (ID 498) [16]	2001	Málaga	12–21	Both	Cross- sectional	Q-EDD/H	Interview semi-structured (Q-EDD) ^A^	⚥ 1555	⚥ 428	⚥ 27.5	⚥ 53	♀ 4.9 ♂ 1.1 ⚥ 3.40	⚥ 7	⚥ 0.5	⚥ 7	⚥ 0.5	⚥ 39	⚥ 2.5	N/A	N/A
Raich, R.M., et al. (ID 500) [17]	2001	Barcelona	18.7–19.1	Female	Cross- sectional	EAT-40	N/A	⚥ 334	⚥ 33	♀ 9.90	N/A	N/A	N/A	N/A	N/A	N/A	N/A	N/A	N/A	N/A
Moraleda Barba, S., et al. (ID 292) [18]	2001	Toledo	13–16	Both	Cross- sectional	EAT-40	N/A	⚥ 503	⚥ 85	♀ 11.51 ♂ 9.87 ⚥ 6.79	N/A	N/A	N/A	N/A	N/A	N/A	N/A	N/A	N/A	N/A
						EAT-40 *			⚥ 34	♀ 10.43 ♂ 9.87 ⚥ 16.97										
Rodríguez, A., et al. (ID 297) [19]	2001	Cádiz	14–18	Both	Cross- sectional	Self-assessment questionnaire	N/A	⚥ 491	⚥ 227	⚥ 46.20	N/A	N/A	N/A	N/A	N/A	N/A	N/A	N/A	N/A	N/A
Ballester Ferrando, D., et al. (ID 283) [20]	2002	Girona	14–19	Both	Cross- sectional	EAT-40	N/A	⚥ 955	⚥ 80	♀ 16.3 ♂ 0.4 ⚥ 8.37	N/A	N/A	N/A	N/A	N/A	N/A	N/A	N/A	N/A	N/A
Espina, A., et al. (ID 617) [21]	2002	Guipúzcoa	11–18	Both	Cross- sectional	EAT-40	N/A	⚥ 969	⚥ 67	♀ 7.1 ♂ 2.4 ⚥ 6.90	N/A	N/A	N/A	N/A	N/A	N/A	N/A	N/A	N/A	N/A
Rojo, L., et al. (ID 273) [22]	2002	Valencia	12–18	Both	Cross- sectional	EAT-40	Clinical interview ^A^	⚥ 544	⚥ 60	⚥ 11.20	⚥ 15	♀ 5.17 ♂ 0.77 ⚥ 2.91	⚥ 1	⚥ 0.18	⚥ 1	⚥ 0.18	⚥ 13	⚥ 2.39	N/A	N/A
Galdón Blesa, M.P., et al. (ID 276) [23]	2003	Albacete	45.9 ± 20.5	Female	Cross- sectional	N/A	Clinical interview ^A^	♀ 175	N/A	N/A	♀ 9	♀ 5.30	♀ 0	♀ 0	♀ 9	♀ 5.3	♀ 0	♀ 0	N/A	N/A
Machado P., et al. (ID 584) [24]	2004	Galicia	17–44	Female	Cross- sectional	EDI **	N/A	♀ 595	♀ 194	♀ 32.60	N/A	N/A	N/A	N/A	N/A	N/A	N/A	N/A	N/A	N/A
Gandarillas, A., et al. (ID 267) [25]	2004	Madrid	15–18	Both	Cross- sectional	EDI ***	N/A	⚥ 4334	⚥ 412	♀ 15.3 ♂ 2.2 ⚥ 9.50	N/A	N/A	N/A	N/A	N/A	N/A	N/A	N/A	N/A	N/A
Vega Alonso, A.T., et al. (ID 258) [26]	2005	Castilla y León	12–17	Both	Cross-sectional	EAT-40 ****	N/A	⚥ 2483	⚥ 305	♀ 13.1 ♂ 10.1 ⚥ 11.60	N/A	N/A	N/A	N/A	N/A	N/A	N/A	N/A	N/A	N/A
						EAT-40			⚥ 204	♀ 12.3 ♂ 3.2 ⚥ 7.8										
Toro, J., et al. (ID 254) [27]	2006	Barcelona	11–18	Female	Cross-sectional	EAT-26	N/A	♀ 468	♀ 84	♀ 17.90	N/A	N/A	N/A	N/A	N/A	N/A	N/A	N/A	N/A	N/A
Peláez Fernández, M.A., et al. (ID 590) [28] ^#^	2007	Madrid	12–21	Both	Cross-sectional	EAT-40 * and/or Q-EDE	Interview semi-structured (EDE) ^D^	⚥ 1545	⚥ 332	⚥ 22.58	⚥ 53	⚥ 3.43	⚥ 3	⚥ 0.19	⚥ 22	⚥ 1.42	⚥ 28	⚥ 1.81	N/A	N/A
Muro-Sans, P., et al. (ID 247) [29]	2007	Barcelona	11–17	Both	Cross-sectional	EDI-2	Interview semi-structured (EDE-12) ^A^	⚥ 1092	⚥ 93	♀5.40 ♂3.57 ⚥ 8.52	♀ 13 ♂ 1 ⚥ 14	♀ 13.97 ♂ 1.07 ⚥ 15.05	0	0	0	0	♀ 13 ♂ 1 ⚥ 14	♀ 13.97 ♂ 1.07 ⚥ 15.05	N/A	N/A
Peláez-Fernández, M.A., et al. (ID 459) [30] ^#^	2008	Madrid	14–18	Both	Cross-sectional	EAT-40 *	Interview semi-structured (Q-EDE) ^D^	⚥ 289	Uninformed	Uninformed	♀ 15 ♂ 3 ⚥ 18	♀ 5.19 ♂ 1.03 ⚥ 6.22	0	0	⚥ 6	⚥ 2.07	⚥ 12	⚥ 4.15	N/A	N/A
						EAT-40 * or IMC ≤ 17.5kg/m^2^	Interview semi-structured (EDE-12) ^D^	⚥ 270	⚥ 47	⚥ 7.40	♀ 7 ♂ 1 ⚥ 8	♀ 2.59 ♂ 0.37 ⚥ 2.96	0	0	⚥ 4	⚥ 1.48	⚥ 4	⚥ 1.48	N/A	N/A
Olesti Baiges, M., et al. (ID 237) [31]	2008	Tarragona	12–21	Female	Cross-sectional	EAT-40 *	Clinical interview ^A^	♀ 551	♀ 75	♀13.61	♀ 50	♀ 9.10	♀ 5	♀ 0.9	♀ 16	♀ 2.9	♀ 29	♀ 5.3	N/A	N/A
Jáuregui Lobera, I., et al. (ID 458) [32]	2008	Sevilla	12–18	Both	Cross-sectional	SCOFF	N/A	⚥ 318	⚥ 61	♀ 29.66 ♂ 6.66 ⚥ 22.80	N/A	N/A	N/A	N/A	N/A	N/A	N/A	N/A	N/A	N/A
						EAT-40			⚥ 30	♀ 15.47 ♂ 2.66 ⚥ 9.43										
Ruiz-Lázaro, P. M., et al. (ID 207) [33] ^#^	2010	Zaragoza	12–13	Both	Cross-sectional	EAT-26	Interview semi-structured (SCAN) ^B,D^	⚥ 701	⚥ 63	10.3 ♂ 7.8 ⚥ 9	⚥ 5	⚥ 1.5	⚥ 0	⚥ 0	⚥ 0	⚥ 0	⚥ 5	⚥ 1.5	N/A	N/A
Mateos-Padorno, C., et al. (ID 206) [34]	2010	Las Palmas de Gran Canaria	12–17	Both	Cross-sectional	EAT-40	N/A	⚥ 1364	⚥ 183	♀ 14.5 ♂ 11.6 ⚥ 13.40	N/A	N/A	N/A	N/A	N/A	N/A	N/A	N/A	N/A	N/A
Cancela Carral, J.M., et al. (D199) [35]	2011	Pontevedra	18–22	Female	Cross-sectional	EAT-40 *	N/A	♀ 258	♀ 41	♀ 15.89	N/A	N/A	N/A	N/A	N/A	N/A	N/A	N/A	N/A	N/A
Cruz-Sáez, M.S., et al. (ID 682) [36]	2013	Guipúzcoa, Navarra	16–20	Female	Cross-sectional	EDI-2	N/A	♀ 767	♀ 116	♀ 15.12	N/A	N/A	N/A	N/A	N/A	N/A	N/A	N/A	N/A	N/A
Martínez González, L., et al. (ID 152) [37]	2014	Vigo; Jaén; Salamanca; León y Huelva	17–23	Both	Cross-sectional	SCOFF	N/A	⚥ 1306	⚥ 255	♀ 21.2 ♂ 15.0 ⚥ 19.5	N/A	N/A	N/A	N/A	N/A	N/A	N/A	N/A	N/A	N/A
Álvarez-Malé, M.L., (ID 137) [38] ^#^	2015	Las Palmas de Gran Canaria	12–20	Both	Cross-sectional	EAT-40 *	Interview semi-structured (EDE) ^D^	⚥ 1342	⚥ 368	♀ 33 ♂ 20.6 ⚥ 27.40	⚥ 34	♀ 5.46 ♂ 2.55 ⚥ 4.11	⚥ 2	⚥ 0.19	⚥ 6	⚥ 0.57	⚥ 26	⚥ 3.34	N/A	N/A
Pérez Martin, P.S., et al. (ID 328) [39] ^#^	2021	Guadalajara	14–18	Both	Cross-sectional	EAT-26	Interview semi-structured (EDE-12) ^A^	⚥ 291	⚥ 12	♀ 5.5 ♂ 2.4 ⚥ 4.1	⚥ 36	⚥ 12.4	⚥ 0	⚥ 0	⚥ 10	⚥ 3.4	⚥ 26	⚥ 8.9	N/A	N/A
						SCOFF			⚥ 36	♀ 15.8 ♂ 8.1 ⚥ 12.40										
						Q-EDE			⚥ 33	♀ 13.9 ♂ 8.9 ⚥ 12										

Score legend: EAT-40: cut-off point ≥ 30; EDI: cut-off point ≥ 14; * EDI: cut-off point ≥ 50; EAT-26: cut-off point ≥ 20; Q-EDD/H: uninformed; Q-EDE: uninformed; EDI-2: cut-off point ≥ 95% on two of the three EDI-2 scales (thinness, bulimia and body dissatisfaction); SCOFF: cut-off point ≥ 2; * EAT-40: cut-off point ≥ 20; ** EDI: cut-off point ≥ 43; *** EDI: combination proposed by authors; **** EAT-40: cut-off point 21–30. Diagnostic legend: ^A^ DSM-IV: Diagnostic and Statistical Manual of Mental Disorders; ^B^ ICD-10: International Classification of Diseases; ^C^ DSM-III-R: Diagnostic and Statistical Manual of Mental Disorders; ^D^ DSM-IV-TR: DSM-III-R: Diagnostic and Statistical Manual of Mental Disorders; ^#^ Tests for Eating Disorders in Controls (false negatives). Tools legend: EDI: Eating Disorder Inventory; EAT: Eating Attitude Test; SCOFF: Sick, Control, One, Fat, Food Questionnaire; Q-EDE: Questionnaire for Eating Disorder Diagnoses; Q-EDE/H: Questionnaire for Eating Disorder Diagnoses for students; EDE: Eating Disorder Examination; SCAN: Schedules for Clinical Assessment in Neuropsychiatry. Other legends: EDNOS: Eating Disorders Not Otherwise Specified; ID: identifier study; N/A: not applicable.

**Table 3 nutrients-16-01513-t003:** Description of studies analysing the prevalence of eating disorder periods in Spain.

Study (ID)	Year	Region	Age	Sex	Type of Study		Tools Screening	Tools Diagnosis	*n*	Total *n* Prevalence According to Tools	%	Total N Prevalence According to Diagnosis	%	*n* Anorexia	%	*n* Bulimia	%	*n* EDNOS	%	*n* Binge Eating Disorder	%
Canals, J., et al. (ID 317) [40]	1997	Tarragona	17.5–18.5	Both	Cohort (follow-up for 60 months)		N/A	Interview semi-structured (SCAN) ^A,B^	⚥ 290	N/A	N/A	⚥ 8 ^A^ ⚥ 4 ^B^	⚥ 2.8 ⚥ 1.3	⚥ 4 ⚥ 1	⚥ 1.4 ⚥ 0.3	⚥ 2 ⚥ 1	⚥ 0.7 ⚥ 0.3	⚥ 2 ⚥ 2	⚥ 0.7 ⚥ 0.7	N/A	N/A
Martínez-González, M.A., et al. (ID 489) [41] ^#^	2003	Navarra	12–21	Females	Cohort (in two stages)	1st stage	EAT-40	Clinical interview ^C^	⚥ 2862	Uninformed	Uninformed	⚥ 119	⚥ 4.15	⚥ 9	⚥ 0.31	⚥ 22	⚥ 0.76	⚥ 88	⚥ 3.07	N/A	N/A
						2nd stage (18 months later)	EAT-40 *	Clinical interview ^C^	⚥ 2509	⚥ 446	⚥ 17.77	⚥ 99	⚥ 3.94	No information	No information	No information	No information	No information	No information	N/A	N/A
Beato-Fernández, L., et al. (ID 480) [42] ^#^	2004	Ciudad Real	12–13	Both	Cohort (in two stages)	1st stage	EAT-40	N/A	⚥ 1076	♀ 73 ♂ 15 ⚥ 88	♀ 6.78 ♂ 1.39 ⚥ 8.17	N/A	N/A	N/A	N/A	N/A	N/A	N/A	N/A	N/A	N/A
							BITE			♀ 67 ♂ 23 ⚥ 90	♀ 6.22 ♂ 2.13 ⚥ 8.36										
						2nd stage (24 months later)	EAT-40	Interview semi-structured (SCAN) ^A, B^	⚥ 1076	♀ 63 ♂ 9 ⚥ 72	♀ 5.85 ♂ 0.83 ⚥ 6.69	⚥ 40	♀ 3.53 ♂ 0.27 ⚥ 3.71	⚥ 1	⚥ 0.1	⚥ 8	⚥ 0.75	⚥ 31	⚥ 2.88	N/A	N/A
							BITE			♀ 71 ♂ 23 ⚥ 94	♀ 6.59 ♂ 2.13 ⚥ 8.73										
Rodríguez-Cano, T., et al. (ID 266) [43] ^#^	2005	Ciudad Real	13	Both	Cohort (in two stages)	1st stage	EAT-40	N/A	⚥ 1766	⚥ 173	♀ 13.10 ♂ 3.00 ⚥ 16.07	N/A	N/A	N/A	N/A	N/A	N/A	N/A	N/A	N/A	N/A
							BITE			⚥ 180	♀ 12.00 ♂ 4.80 ⚥ 16.72										
						2nd stage (24 months later)	EAT-40	Interview semi-structured (SCAN) ^C^	⚥ 1076	⚥ 159	♀11.3 ♂ 1.8	⚥ 40	⚥ 3.71	⚥ 1	⚥ 0.17	⚥ 8	⚥ 0.75	⚥ 31	⚥ 2.88	N/A	N/A
							BITE				♀ 12.7 ♂ 3.1										
Sancho, C., et al. (ID 241) [44] ^#^	2007	Tarragona	9–13.5	Both	Cohort (in two stages)	1st stage	ChEAT	Interview semi-structured (DICA-C) ^C^	⚥ 1336	♀ 94 ♂ 79 ⚥ 173	♀ 13.68 ♂ 12.17 ⚥ 12.94	⚥ 46	⚥ 3.44	⚥ 0	⚥0	⚥ 0	⚥ 0	⚥ 45	⚥ 3.36	⚥ 1	⚥ 0.07
						2nd stage (24 months later)	EAT y/o BITE	Interview semi-structured (DICA-A) ^C^		⚥ 113	⚥ 8.45	⚥ 68	⚥ 5.08	⚥ 0	⚥ 0	⚥ 5	⚥ 0.37	⚥ 59	⚥ 4.41	⚥ 4	⚥ 0.29
Rojo-Moreno, L., et al. (ID 141) [45]	2015	Valencia	12–16	Both	Cohort (in two stages)	1st stage	Uninformed	Interview semi-structured (K-SADS) ^C^	⚥ 962	Uninformed	Uninformed	⚥ 35	⚥ 3.6	⚥ 2	⚥ 0.20	⚥ 3	⚥ 0.31	⚥ 30	⚥ 3.11	⚥ 0	⚥ 0
						2nd stage (24 months later)			⚥ 326			⚥ 12	⚥ 3.68	⚥ 8	⚥ 2.45	⚥ 3	⚥ 0.92	⚥ 1	⚥ 0.30	⚥ 0	⚥ 0
Esteban-Gonzalo, L., et al. (ID 356) [46]	2019	Madrid	11–19	Both	Cohort	1st stage	SCOFF	N/A	⚥ 847	♀ 97 ♂ 60 ⚥ 157	♀ 23.6 ♂ 13.8 ⚥ 18.53	N/A	N/A	N/A	N/A	N/A	N/A	N/A	N/A	N/A	N/A
						2nd stage (24 months later)	SCOFF		⚥ 677	♀ 32 ♂ 11 ⚥ 43	♀ 9.7 ♂ 3.2 ⚥ 6.35										

Score legend: EAT-40: cut-off point ≥ 30; BITE: cut-off point ≥ 10; * EAT-40: cut-off point ≥ 21; ChEAT: cut-off point ≥ 17; EAT: cut-off point ≥ 25; SCOFF: cut-off point ≥ 2. Tools legend: BITE: Bulimic Investigatory Test Edinburgh; ChEAT: Children Eating Attitudes Test; DICA: Diagnostic interview for children and adolescents; DICA-A: Diagnostic interview for adolescents; DICA-C: Diagnostic interview for children; EAT: Eating Attitude Test; K-SADS: Kiddie Schedule for Affective Disorders and Schizophrenia; SCAN: Schedules for Clinical Assessment in Neuropsychiatry; SCOFF: Sick, Control, One, Fat, Food Questionnaire. Diagnostic legend: ^A^ ICD-10: International Classification of Diseases; ^B^ DSM-III-R: Diagnostic and Statistical Manual of Mental Disorders; ^C^ DSM-IV: Diagnostic and Statistical Manual of Mental Disorders; ^#^ Tests for Eating Disorders in Controls (false negatives). Other legends: EDNOS: Eating Disorders Not Otherwise Specified; ID: identifier study; N/A: not applicable.

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
