# Peer review of "A Scoping Review of the Prevalence of Eating Disorders in Spain"

_nutrients, 2024, doi:10.3390/nu16101513_

Round 1

Reviewer 1 Report

Comments and Suggestions for Authors

This is a review that aims to analyze the prevalence of Eating Disorders in Spain.                                                                                                             This is good research, but is not suitable for publication in a nutrition journal like Nutrients.                                                                                                    It seems to be more suitable for journals dealing with Eating Disorders or better yet, Spanish public health.

- Excluding obesity from the inclusion criteria is a mistake because many of the Eating Disorders cause obesity, and in this way the Studies to be selected for the review would greatly increase.

- The selected Studies collect patients aged between 11 and 23 years: it makes no sense to indicate < 65 years in the text.

- The Authors, very correctly, declare that this review demonstrates an extreme heterogeneity in the ways of classifying Eating Disorders, therefore the conclusions do not bring new results either.

- No selected work uses the criteria proposed by the DSM-5.

- References 63,64,65 are not cited in the text.

Author Response

Dear reviewer,

Thank you very much for your constructive feedback. The corresponding authors appreciate the remarks and insights provided on our work. We will address the subsequent questions and concerns raised in your review. All changes have been included in the document, along with change control.

Reviewer 1.

This is a review that aims to analyze the prevalence of Eating Disorders in Spain.  This is good research, but is not suitable for publication in a nutrition journal like Nutrients. It seems to be more suitable for journals dealing with Eating Disorders or better yet, Spanish public health.

  • Thank you very much for your input. We agree with your perspective, but we also believe that the journal's Nutritional Epidemiology section is a perfect fit for the aim of our research. We chose to include it there because. In fact, we sent it because we consider mental health as they reflect: "The Nutritional Epidemiology section aims to publish contributions on the impact of food consumption and nutritional status on different aspects of health. Papers should contribute new knowledge, from descriptive cross-sectional studies to those that contribute to the prevention of nutrition-related diseases such as obesity, type 2 diabetes, cancer, osteoporosis and cardiovascular diseases and other conditions such as frailty or mental health".

Excluding obesity from the inclusion criteria is a mistake because many of the Eating Disorders cause obesity, and in this way the Studies to be selected for the review would greatly increase.

  • We acknowledge the connection between eating disorders and obesity. Nevertheless, our goal was to conduct the inaugural review of eating disorders within a healthy population in Spain, specifically excluding any associated pathologies. It is important to note this the first review of its kind to be carried out in Spain, focusing on a population without pre-existing health conditions.

The selected Studies collect patients aged between 11 and 23 years: it makes no sense to indicate < 65 years in the text.

  • We specified an age criterion of under 65 for the scientific search to ensure a comprehensive review of potential articles. However, the studies ultimately selected for inclusion only involved participants between 11 and 23 years.

The Authors, very correctly, declare that this review demonstrates an extreme heterogeneity in the ways of classifying Eating Disorders, therefore the conclusions do not bring new results either.

  • We acknowledge the heterogeneity in classifying eating disorders highlighted in this review. However, we consider this review to be valuable as it is the first conducted in Spain. Furthermore, the diversity among the included articles highlights the necessity for more standardized approaches in future research.

No selected work uses the criteria proposed by the DSM-5.

  • Indeed, our search revealed no studies that analyzed samples using the DSM-5 criteria. We have noted this in our review, highlighting the lack of published articles that meet these specific criteria within our search parameters.

References 63, 64, 65 are not cited in the text.

  • We regret this oversight. References 63, 64 and 65 have been removed, and the literature has been updated accordingly.

Reviewer 2 Report

Comments and Suggestions for Authors

Introduction needs to include more informastion regarding the eating disorders and the Manual of Mental Disorders in its 5th revision snd the In- 38 ternational Classification of Diseases (ICD-11) . The result section is sufficient. However further elaboration can be added to enrich the results found in the discussin section.

Addition of diagrams and further examples can also be included

----

1. What is the main question addressed by the research?

comments: Yes, however more information is required to fulfill the highest level of relevance.

2. What parts do you consider original or relevant for the field?

comments: That their is no review of indexed publications studying their prevalence in Spain is available up to this day.

2.1 What specific gap in the field does the paper address?

comments: The prevalence of eating disorders in spain using a scoping review search.

3. What does it add to the subject area compared with other published

material?

comments: The degree of uniqueness is not that high as the research doesn’t bring something extremely different from other present studies.

4. What specific improvements should the authors consider regarding the

methodology?

comments: The methodology is okay

4.1 What further controls should be considered?

comments: No further controls

5. Please describe how the conclusions are or are not consistent with the

evidence and arguments presented.

comments: The conclusion is consistent but can be improved in terms of depth and on to the point key.

5.1 Please also indicate if all main questions posed were addressed and by which specific experiments.

comments: Yes but further elaboration is required.

6. Are the references appropriate?

comments: Yes.

7. Please include any additional comments on the tables and figures and

quality of the data.

comments: The figures added are adequate.

Author Response

Dear reviewer,

Thank you very much for your constructive feedback. The corresponding authors appreciate the remarks and insights provided on our work. We will address the subsequent questions and concerns raised in your review. All changes have been included in the document, along with change control.

Reviewer 2.

Introduction needs to include more information regarding the eating disorders and the Manual of Mental Disorders in its 5th revision and the 38 International Classification of Diseases (ICD-11). The result section is sufficient. However further elaboration can be added to enrich the results found in the discussion section.

  • In the second paragraph of the introduction we include precise information on the diagnoses according to DSM-5 and ICD-11. In addition, due to changes compared to DSM-IV, we specify that new diagnostic criteria have been introduced for some pathologies. However, if you believe that the inclusion of further information is necessary, we are more than willing to incorporate it.

Addition of diagrams and further examples can also be included.

  • The flowchart is presented in figure 1, developed in accordance to the PRISMA recommendations. Should you suggest the creation of an additional diagram, we are prepared to do so.
  1. What is the main question addressed by the research?

comments: Yes, however more information is required to fulfill the highest level of relevance.

  • We appreciate your feedback. We are committed to maintaining the highest quality standards and will gladly provide additional information to enhance the relevance of our study.
  1. What parts do you consider original or relevant for the field?

comments: That there is no review of indexed publications studying their prevalence in Spain is available up to this day.

  • Thank you very much.

2.1 What specific gap in the field does the paper address?

comments: The prevalence of eating disorders in spain using a scoping review search.

  • That's right.
  1. What does it add to the subject area compared with other published material?

comments: The degree of uniqueness is not that high as the research doesn’t bring something extremely different from other present studies.

  • We acknowledge your concerns about the uniqueness of our study. We believe its relevance lies in being the first comprehensive review of this topic conducted in Spain, which addresses a significant gap in regional literature."
  1. What specific improvements should the authors consider regarding the methodology?

comments: The methodology is okay

  • Thank you very much.

4.1 What further controls should be considered?

comments: No further controls

  • Indeed, as it is a revision, it has no controls.
  1. Please describe how the conclusions are or are not consistent with the evidence and arguments presented.

comments: The conclusion is consistent but can be improved in terms of depth and on to the point key.

  • Thank you for your feedback. We agree that our conclusions are consistent with the evidence presented, although they could benefit from greater depth and clarity.

5.1 Please also indicate if all main questions posed were addressed and by which specific experiments.

comments: Yes but further elaboration is required.

  • Thank you very much.
  1. Are the references appropriate?

comments: Yes.

  • Thank you very much.
  1. Please include any additional comments on the tables and figures and quality of the data.

comments: The figures added are adequate.

  • Thank you very much.

Round 2

Reviewer 1 Report

Comments and Suggestions for Authors

The Authors have improved the paper and have better explained their point of view